# Individualized Multimodal Immunotherapy for Adults with IDH1 Wild-Type GBM: A Single Institute Experience

**DOI:** 10.3390/cancers15041194

**Published:** 2023-02-13

**Authors:** Stefaan W. Van Gool, Jennifer Makalowski, Peter Van de Vliet, Stefanie Van Gool, Tobias Sprenger, Volker Schirrmacher, Wilfried Stuecker

**Affiliations:** Immun-Onkologisches Zentrum Köln, 50674 Cologne, Germany

**Keywords:** glioblastoma, immunotherapy, dendritic cell vaccine, oncolytic virus, modulated electrohyperthermia

## Abstract

**Simple Summary:**

The standard of care for patients with primary glioblastoma multiforme consists of neurosurgery, radiochemotherapy, and maintenance chemotherapy. We added individualized multimodal immunotherapy to this treatment. During maintenance chemotherapy, immunogenic cell death immunotherapy (oncolytic virus injections and sessions of modulated electrohyperthermia) was inserted. After chemotherapy, active specific immunotherapy with dendritic cell vaccines (IO-Vac^®^) and modulatory immunotherapy were given. The manuscript describes the retrospective analysis of a group of 50 adults taken out of the database following a predefined clinical profile without further bias. We observed a clearly improved overall survival in comparison to published data. There were no major adverse reactions. The proposed treatment concept takes into account dynamic changes in tumor biology and tumor–host interaction, proposing an additional perspective besides the paradigm of protocol medicine in clinical trials. This report on real-world data has great scientific value and high relevance for patients.

**Abstract:**

Synergistic activity between maintenance temozolomide (TMZm) and individualized multimodal immunotherapy (IMI) during/after first-line treatment has been suggested to improve the overall survival (OS) of adults with IDH1 wild-type MGMT promoter-unmethylated (unmeth) GBM. We expand the data and include the OS of MGMT promoter-methylated (meth) adults with GBM. Unmeth (10 f, 18 m) and meth (12 f, 10 m) patients treated between 27 May 2015 and 1 January 2022 were analyzed retrospectively. There were no differences in age (median: 48 y) or Karnofsky performance index (median: 80). The IMI consisted of 5-day immunogenic cell death (ICD) therapies during TMZm: Newcastle disease virus (NDV) bolus injections and sessions of modulated electrohyperthermia (mEHT); subsequent active specific immunotherapy: dendritic cell (DC) vaccines plus modulatory immunotherapy; and maintenance ICD therapy. There were no differences in the number of vaccines (median: 2), total number of DCs (median: 25.6 × 10^6^), number of NDV injections (median: 31), and number of mEHT sessions (median: 28) between both groups. The median OS of 28 unmeth patients was 22 m (2y-OS: 39%), confirming previous results. OS of 22 meth patients was significantly better (*p* = 0.0414) with 38 m (2y-OS: 81%). There were no major treatment-related adverse reactions. The addition of IMI during/after standard of care should be prospectively explored.

## 1. Introduction

Since the wide implementation of checkpoint inhibitors (CPIs) in oncologic treatments, the term immunotherapy became a key term in neuro-oncology for the treatment of glioblastoma multiforme (GBM) [1]. Some years later, however, immunotherapy for the treatment of GBM did not fulfill expectations. Many papers have reviewed the challenges [2,3,4]. One of the key issues is the use of the term immunotherapy itself. The American Cancer Society defines immunotherapy as “a treatment that uses a person’s own immune system to fight cancer. Immunotherapy can boost or change how the immune system works so it can find and attack cancer cells” [5]. A prerequisite for immunotherapy is the existence of tumor antigens expressed in MHC molecules, making the tumor cells a target for the immune cells that can recognize them via the T-cell receptor. Alternatively, a lack of MHC molecule expression can make the tumor cells prone to NK cell recognition. A second prerequisite for immunotherapy is a functioning immune system.

Over the decennia, several types of immunotherapy have been elaborated, and many of them were investigated to treat patients with GBM [6].

1. In restorative immunotherapy, cytokines are administered to the patient with the aim to provide stimulatory signals, thereby changing immune balances. This form of immunotherapy is not specific and causes a lot of side effects, at the end without clear benefit [7]. Restorative immunotherapy has been abandoned in the domain of neuro-oncology.

2. The administration of antibodies that can bind to specific antigens and elicit complement-mediated or antibody-dependent cell-mediated cytotoxicity is named passive immunotherapy [8]. This form of immunotherapy is not well elaborated for GBM because of the existence of the blood–brain barrier, making penetration of the antibody into the tumor microenvironment almost impossible. Moreover, the antigen targets of the antibodies are usually not homogeneously present in all tumor cells, hampering the induction of tumor control. Nevertheless, some antibodies have been used in the context of GBM, e.g., for targeting radioactive compounds in the tumor, thereby killing tumor cells [9].

3. Adoptive T-cell therapy consists of T cells that are expanded and manipulated *ex vivo*, then infused back into the body with the aim to reach the tumor. The use of lymphokine-activated killer (LAK) cells or tumor-infiltrating lymphocytes (TILs) has not been extensively explored in the domain of GBM, in contrast to other cancer types such as melanoma. Novel biotechnology approaches reinforced the adoptive T-cell immunotherapy strategy. Nowadays, CAR T cells and TCR-transduced T cells are available, targeting tumor-specific or tumor-associated antigens [10,11,12]. Adoptive T-cell therapy using expanded CMV-specific T cells is a novel and interesting approach [13]. A lot of preclinical and clinical research is going on to explore the efficacy of these advanced therapy medicinal products (ATMPs) for the treatment of patients with GBM.

4. A more novel type of immunotherapy, immunogenic cell death (ICD) immunotherapy, has recently entered the clinical field. The hallmarks of ICD have been defined in the literature [14,15]. ICD immunotherapy is aimed to induce tumor cell death in such a mode that danger signals are elicited, and as a consequence, an anticancer immune response is generated within the body. Several old and new treatment strategies might be explored to elicit ICD, among which radiotherapy is the oldest. Although chemotherapy can elicit ICD, most of the drugs with such a working mechanism are not used to treat GBM. The administration of oncolytic viruses, such as Newcastle disease virus (NDV), is a novel treatment modality, which clearly can induce ICD [16]. Finally, the application of locoregional, nonionizing electromagnetic waves with the aim to control GBM is based on the induction of ICD. Two applications are available in clinical reality for patients with GBM. Tumor-treating fields (TTFs), based on electromagnetic waves in the kilohertz range (100 to 300 kHz) at low intensity (1 to 3 V/cm), are applied at least 85% of the time [17,18]. The original working mechanism was supposed to be the cell cycle arrest, but it became clear that a strong immune component is present as well [19]. The technology of modulated electrohyperthermia (mEHT) consists of electromagnetic waves in the megahertz range (13.56 MHz), and can repetitively be applied with a power of 40 to 150 watts from 20 to 60 min [20]. The ICD working mechanism of modulated electrohyperthermia has been demonstrated in preclinical models and has been observed in patients [21,22].

5. With the use of active specific immunotherapy, the aim is to actively stimulate the body to build up an immune response against the aimed antigen [23,24]. The critical components of active specific immunotherapy are the antigens, the vehicle to bring the antigens into the body, and most importantly the necessary danger signal. The antigens can be derived from DNA, RNA, proteins, or peptides. They can be tumor-specific, eventually in combination, or they can cover all potential known and unknown tumor antigens. The vehicle is as important as the antigen. The antigen should reach the lymph nodes, and the danger signals must be present to be able to induce an antitumor antigen-specific T-cell response. Dendritic cells (DCs) are the most efficient antigen-presenting cells to induce an immune response. For this, patient-derived blood cells, monocytes or circulating DCs, are cultured *ex vivo* for loading with tumor antigens and subsequent maturation with cytokine mixtures with or without Toll-like receptor stimulators, which reflect the danger signals. These ATMPs are then injected back into the patient, mostly via the intradermal and sometimes intranodal route. For some vaccines, the antigen is injected into the patient, and danger signals are separately elicited with drugs such as local application with imiquimod or coinjection with GM-CSF.

6. A final modality of immunotherapy is modulatory immunotherapy. With this type of immunotherapy, an already-existing anticancer immune response is modulated. With checkpoint inhibitors (CPIs) such as anti-PD1 or anti-PDL1, a blocking signal from tumor cells to activated anticancer cytotoxic T lymphocytes (CTLs) is blocked, and hence, the anticancer CTL functioning is deblocked. Because a spontaneous anti-GBM immune response, such as in cases of hypermutant GBM, is rarely elicited, there is no rationale to explore the efficacy of CPIs as a single immunotherapy strategy for GBM, certainly not if even the immune functioning as such is further inhibited due to chemotherapy or use of steroids [25]. Meanwhile, several checkpoints of the anticancer immune response are known, and novel CPIs are under development. Other types of modulatory immunotherapy for GBM are the metronomic cyclophosphamide to deplete regulatory T cells, or metronomic capecitabine to deplete myeloid-derived suppressor cells (MDSCs).

Most of the immunotherapy modalities used at IOZK are patient-specific. Whereas the term “personalized” reflects rather the specific characteristics of the tumor cell and its specific biologic features, “individualized” reflects more the patient as a whole entity having a unique immune system, a unique tumor, and a unique tumor–host interaction. Because each patient is treated with combinations of immunotherapy modalities, which is now even suggested in the literature [26], the term “Individualized multimodal immunotherapy (IMI) was initiated in 2020 [27].

The implementation of novel treatment technologies in clinical practice usually occurs at the time of advanced or relapsing disease when standard therapies are not possible anymore. Because ICD immunotherapy, active specific immunotherapy, and modulatory immunotherapy require a healthy immune system in the body to build up and/or facilitate the anticancer immune response, these immunotherapy modalities are at best connected to the first-line treatment, when the tumor burden is maximally reduced by the standard oncologic treatments. This concept created a new challenge in how to design the most optimal smart combination of standard of care and immunotherapy. In the HGG-2010 trial (EudraCT 2009-018228-14), DC vaccines were placed after the radiochemotherapy with boost vaccines during the maintenance temozolomide (TMZm) treatment *versus* DC and boost vaccines after chemotherapy. This crossover trial design allowed a double-blinded, prospective, randomized, placebo-controlled primary readout of progression-free survival (PFS) in patients with vaccination *versus* placebo after 6 courses of TMZm, and a secondary unblinded readout of OS of the patients with vaccination during *versus* after chemotherapy. The trial was prematurely closed, and PFS data were not published, but a prolongation of OS for all molecular subtypes of GBM due to DC vaccination was obvious [28]. In both complete resected and less-than-complete resected patients, the 2-year OS tended to be higher in the patients receiving DC vaccines after TMZm. Therefore, we considered that active specific immunotherapy with DC vaccines (IO-Vac^®^) should be kept after the TMZm period. In order to strengthen the tumor control during the TMZm monotherapy, NDV injections and sessions of mEHT were added as ICD immunotherapy shortly after each 5-day chemotherapy block, aimed to kill the GBM cells with a second cell killing modality (ICD) different from the chemotherapy (alkylating agent). Additional ICD immunotherapy blocks were maintained after two IO-Vac^®^ DC vaccines to maintain the anticancer immune responsiveness. With this combined treatment approach, we found strong suggestions for synergistic activity between the standard-of-care chemotherapy and the immunotherapies to improve the OS of patients with IDH1 wild-type MGMT promoter unmethylated adults with GBM [29]. In the current retrospective analysis of our patients, we report the clinical and laboratory data of IDH1 wild-type MGMT promoter-methylated and -unmethylated adults with GBM.

## 2. Patients and Methods

### 2.1. Patients

IOZK obtained approval on 27 May 2015 to produce IO-Vac^®^: “specific autologous anti-tumor dendritic cell vaccine for intradermal application: Patient-autologous monocyte-derived dendritic cell, loaded with tumor antigens from lysates from autologous tumor cells and matured with danger signals from Newcastle Disease Virus” (DE_NW_04_MIA_2015_0033 and DE_NW_04_MIA_2020_0017). This ATMP can be used to treat patients within the legal framework of individualized treatment (“individueller Heilversuch”). Informed consent was to be signed prior to the start of individualized treatment.

All patients treated between 27 May 2015 and 1 January 2022 were taken into the data set, with a further observation period for surviving patients extended to 1 July 2022. A retrospective analysis of these patients was performed. There were 218 patients with a diagnosis of GBM. We next selected the GBM patients older than 18 years with a documented IDH1 status. There were 110 patients with IDH1 status documented as wild type. We finally focused on the patients with the first event of IDH1 wild-type GBM and documented methylation status of the MGMT promoter. The categorization of the patients as MGMT promoter methylated or unmethylated was based on the available pathology report. There were 22 patients with MGMT promoter-methylated and 29 patients with MGMT promoter-unmethylated status. In the latter group, one patient was lost for follow-up. The final study population therefore consisted of 50 patients.

All patients received individualized multimodal immunotherapy at one place (www.iozk.de (accessed on 1 September 2022)) while the standard of care was provided in the local neuro-oncology clinic. Patients came from 16 different countries to receive IMI in Cologne, Germany. When IMI was administered no longer, follow-up of the patients was organized via phone or mailing.

### 2.2. Individualized Multimodal Immunotherapy

Several modes of immunotherapy were provided. ICD immunotherapy consisted of injections of Newcastle disease virus (NDV) in combination with sessions of mEHT. The mesogenic oncolytic MTH-68 strain NDV was used. One dose consisted of 1 to 10 × 10^7^ infectious particles. Whereas the NDV was administered in the earlier years as a short infusion diluted in NaCl 0.9%, the administration was switched to a bolus injection in 2018. In 2021, we decided to include 100 mL of mannitol 10% infusion prior to each NDV bolus injection. The mEHT was administered with the use of the Oncothermia EHY-2000 device (www.Oncotherm.com (accessed on 1 September 2022)). The dose per session was 40 watts for 50 min. Since 2018, the dose was allowed to be increased to 60 (80) watts, if tolerated. Nevertheless, most patients were kept on 40-watt treatment for most of the sessions. During each mEHT session, infusions containing 7.5 g of vitamin C, 40 mg of MgCl2, 45 mg of CaCl2, 15 mg of KCl, 10 mg of Magnesiocard, 5 mL of Nervoregin, and eventually Selenase (depending on the blood level) were administered [30]. In 2018 we also increased the ICD course intensity from 3 daily NDV/mEHT sessions to 5 daily NDV/mEHT sessions.

The active specific immunotherapy was performed with the dendritic cell vaccine IO-Vac^®^. This vaccine consists of autologous mature DCs, differentiated out of peripheral blood monocytes in the presence of IL-4 and GM-CSF, loaded with tumor antigens, and matured in the presence of the cytokine cocktails IL-1β, IL-6, and TNF-α, together with NDV as described [30]. The source of tumor antigens was proteins. These were derived from lysates created ex vivo from snap-frozen tumor tissue. Alternatively, serum was obtained after 5 days of ICD immunotherapy and was used for loading the immature DCs. In this method, the ICD immunotherapy-induced increased concentration of antigenic extracellular microvesicles and apoptotic bodies (summarized here as extracellular vesicles or EVs) reflects an in vivo lysate of residual tumor cells. The vaccine was administered intradermally in the upper third of the arm.

The modulatory Immunotherapy was tailored for each patient, based on the results of the immune-diagnostic blood samples obtained prior to the start of the IMI.

### 2.3. Accompanying Laboratory Tests

Routine laboratory tests were performed by the local laboratory for ambulant medical care in Nordrhein Westfalen (www.Laborunion.de (accessed on 1 September 2022)). Immunology laboratory tests were performed at Synlab (www.synlab.de (accessed on 1 September 2022)). Part of the immunology laboratory tests was performed in the immune-diagnostic laboratory of the IOZK. During the more than six-year observation period covered in this retrospective analysis, test methods and locations changed. For each test at each moment, the result was compared to the available normal range, and categorized as lower, within, or higher than the normal range values. Circulating cancer cells (CCCs) and chemosensitivity tests (including mRNA expression for PDL1 and MGMT) were performed at the Molecular Oncology Department of Biofocus (www.biofocus.de/molecular-oncology (accessed on 1 September 2022)). The chemosensitivity tests for CCCs were considered meaningful when the ratio of mRNA expression of the target gene was higher than 2 in comparison to the mRNA expression for the GAPDH housekeeping gene. High mRNA for PDL1 is interpreted as CCCs that have the potency to express PDL1 as a defense against an immune attack, which can be counteracted with checkpoint inhibitors. High mRNA expression for MGMT is interpreted as CCCs that have the potency for MGMT enzyme activity and DNA repair of DNA damage after radiotherapy and/or chemotherapy.

### 2.4. Statistics

All statistical tests were performed using GraphPad Prism version 7.01 for Windows, GraphPad Software, La Jolla, CA, USA, www.graphpad.com.

## 3. Results

### 3.1. Patient Characteristics

The total of 50 analyzed adults with GBM (23 females, 27 males) were all registered as IDH1 wild type, and were divided according to the registered MGMT promoter methylation status: 22 patients (12 females, 10 males) had methylated, and 28 patients (11 females, 17 males) had unmethylated MGMT promoter (Figure 1). There was no significant difference in age (Mann–Whitney test). The median age was, respectively, 54 (ranging from 26 to 72) and 47 years (ranging from 18 to 65). In addition, the registered Karnofsky performance index (KPI) was equally distributed (Mann–Whitney test). Median KPI was 80 in both groups, ranging in the methylated patients from 60 to 100 and in the unmethylated patients from 50 to 100. The extent of resection did not differ between both groups (Chi-square test): complete resection in 5 methylated and 9 unmethylated patients; less-than-complete resection in 14 and 12 patients, respectively; and not documented in 3 and 7 patients, respectively. From these basic characteristics, we concluded that both patient groups were comparable at the level of the clinical risk profile.

Patients started standard-of-care treatment, being neurosurgical resection, radiochemotherapy, and eventually maintenance chemotherapy with temozolomide (TMZ). At a certain time point during this first-line treatment, patients presented at IOZK for additional IMI integrated within and/or added to the first-line treatment. All patients started with an immune-oncologic evaluation and an immune-diagnostic blood sampling before the start of IMI. The time between the operation and immune-diagnostic procedures was a median of 3 months for both groups ranging from 0.5 to 13 months (Figure 1). Data for the different blood variables were registered in 36 to 50 out of the 50 patients. The results of the immune-diagnostic blood tests are summarized in Table 1. In both MGMT promoter-methylated and MGMT promoter-unmethylated patients, the distribution (low, normal, high) of the different variables of blood cells, in particular immune cells, was similar. A relatively higher percentage of patients with MGMT promoter-methylated tumors had a low total white blood cell count (Chi-square test: *p* = 0.04). We therefore concluded that the basic blood counts and lymphocyte counts were comparable in both MGMT promoter-methylated and -unmethylated patients. In both groups, a substantial part of the patients was lymphopenic (32/49) and had low platelet counts (15/50). The proportion of patients with a low number of T cells (37/48), B cells (38/48), and NK cells (32/48) was remarkably high. Th17 skewing was detected in a lot of patients (15/36), while Th1, Th2, and their ratio were mostly within normal ranges (38/50 and 31/50, respectively). The NK cell function was low in a large proportion of patients (34/49). The oxidative stress was moderately or clearly increased in 34/46 patients. Most patients (33/46) showed a high total antioxidative capacity.

Blood was also taken for the detection of CCCs. There was no difference in the distribution of patients without CCCs, patients with CCCs but negative for PDL1 mRNA expression, and patients with CCCs and positive for PDL1 mRNA expression (Table 1). The detection for expression of PDL1 on tumor cells in pathology was not available, except for Patient 24762, who had on pathology 60% of tumor cells being positive for PDL1.

During the 6.5-year observation period, the MGMT mRNA expression analysis became available as a chemosensitivity test for TMZ, and became included in the immune-diagnostic evaluation. Based on the pathology report, 22 patients were registered as MGMT promoter-methylated. We analyzed the data within the first 4 months after surgery. In 17 patients, the mRNA for MGMT was not measured because the blood sampling was not yet performed, because there were no CCCs detected, or because the MGMT analysis for mRNA expression on CCCs was not available. CCCs were categorized as low for mRNA for MGMT in 3 patients, and high in 2 patients. For the 28 MGMT promoter-unmethylated patients, there were no measurements available within the first 4 months in 21 patients, while 4 patients showed low RNA expression and 3 patients showed high RNA expression for MGMT in their CCCs.

### 3.2. Treatment

Neurosurgery, radiochemotherapy, maintenance chemotherapy, imaging, revalidation, and supportive care were always performed by the multidisciplinary neuro-oncology team where the patient started diagnosis and treatment. All immunotherapy treatments were performed at the IOZK in Cologne, Germany, in a daycare facility. The time between the operation and the first treatment day of IMI was for both groups a median of 4 months, ranging from 2 to 13 months (Figure 1). Patients could be divided into three treatment subgroups: IO-Vac^®^ treatment was started after local therapy, without TMZ maintenance chemotherapy (IT Start 1); TMZ maintenance chemotherapy was accompanied by ICD immunotherapy, and IO-Vac^®^ vaccines were given after all chemotherapy was finished (IT Start 2); IO-Vac^®^ treatment was started only after completion of TMZ maintenance chemotherapy (IT Start 3). The distribution of the patients according to these timelines for the immunotherapy start was again equal for both the MGMT promoter-methylated and -unmethylated patients (Chi-square test): 2, 16, and 4, and 2, 23, and 3 patients, respectively. Hence, 78% of the patients were treated with local therapy, TMZ maintenance plus ICD immunotherapy, IO-Vac^®^ vaccines, and maintenance ICD immunotherapy courses. For all patients, 3- to 5-day ICD immunotherapy courses were continued as maintenance with a frequency individually determined according to the feasibility for each patient, the residual tumor disease, and the observed response to treatment.

Figure 2 shows the details for ICD and active specific immunotherapy for both MGMT promoter-methylated and -unmethylated patients. The number of NDV administrations, the number of sessions of mEHT, the number of vaccinations, and the total number of DCs administered were not different between both MGMT promoter-methylated and -unmethylated patients. In most patients, the number of NDV injections and the number of mEHT sessions were equal, as they were aimed to be performed together for the induction of ICD. However, in one patient of the methylated group (p23092), and in three patients of the unmethylated group (p23238, p23804, p24163), the sessions of mEHT were less than the injections of NDV. In all cases, this was the personal decision of the patient, which was based on the use of TTFs, and in one case also the use of additional modulated electrohyperthermia in another center (p24163). Two further cases in the MGMT promoter-unmethylated group used TTFs but were treated with the combined ICD immunotherapy, though only 15 (p25281) and 16 (p24478) sessions, respectively, due to fast disease progression.

The source of antigens for loading into the DC vaccine did not differ between MGMT promoter-methylated and -unmethylated patients. In the former group, ICD immunotherapy-induced serum-derived EVs were used as a source of antigen in 17 patients, and tumor lysate (TL) in two patients. Three patients did not receive DC vaccines. In the latter group, 15 patients received DCs loaded with serum-EVs, 1 patient received DCs loaded with TL, and 4 patients received DCs loaded with both EVs and TL. In this group, eight patients did not receive DC vaccines.

One patient (p24542 for whom more details will be described below) in the MGMT promoter-methylated group and three patients (p24373, p24762, 25298) in the promoter-unmethylated group started with anti-PD1 therapy as part of IMI. Two further patients in the MGMT promoter-methylated group (p23346, p24005) received anti-PD1 therapy later during primary treatment, because of the increased mRNA expression of PDL1 in the CCCs during treatment.

During the 6.5-year period covered in this retrospective analysis, some complementary medicine strategies were implemented stepwise. A metabolic treatment consisted of metformin 500 mg q12h and mebendazole100 mg q24h. Great emphasis was given to anti-inflammatory treatment, which consisted of a buccal administration of curcumin and celecoxib 200 mg q12h. The psycho–neuro–endocrine–immunology balance in the brain was influenced by recommending melatonin 20 mg in the evening, and by allowing CBD/THC use to a tolerated dose upon the wish of the patient. Finally, the neuroglioma connection was inhibited with mirtazapine, fluoxetine, or perampanel, according to the tolerance of the patient, and with a personalized dose to avoid accompanying side effects. Mirtazapine was the first recommendation because of its anti-histamine receptor-1 blocking effect. When this drug was not tolerated, loratadine or clemastine was recommended as an anti-histamine receptor-1 blocker.

One particular and interesting patient could not follow the standard of care (Figure 3). Patient 24542 was 34 years old when she was diagnosed with GBM temporal right, IH1wt, MGMT hypermethylated. Hypermutation was present in the context of POLE syndrome. She was treated with an R0 resection and afterwards radiochemotherapy. It was decided by the treating physicians not to initiate TMZm therapy, but Avastin together with pembrolizumab. The treating physicians also requested the IOZK to initiate IMI. In parallel to the treatment with Avastin and pembrolizumab, both given from Months 5 to 21, 22 treatments consisting of 3 to 5 days of NDV injections and sessions of mEHT were given from Months 5 to 26. Two IO-Vac vaccines were given at Months 15 and 16. Thirty months after the first diagnosis, she depicted a multifocal relapse with lesions in the spine. After local radiotherapy, chemotherapy with lomustine was initiated and, again strengthened with ICD immunotherapy. The patient also received repurposing drugs. Nine months later, at the time of writing, she showed a status of remission, and the combined treatment was stopped.

### 3.3. Evolution of the Patients

During treatment, repetitive blood samples were taken for evaluation of the immune function and of the evolution of CCCs. Figure 4 shows the evolution of the Th1, Th2, Th17, and NK cell functions of the patients over time. Th17 analysis was not yet available at the beginning of the period of analysis. Once available, 38% of the MGMT promoter-methylated patients and 43% of the MGMT promoter-unmethylated patients showed a Th17 skewing with values above the expected normal range. In the MGMT promoter-methylated patients, all five patients with high Th17 skewing normalized at a certain time point during treatment. In the MGMT promoter-unmethylated patients, five patients had a follow-up sample, two patients showed a normalization, while three patients remained Th17 high during their treatment. The NK cell function was followed in a similar way. Follow-up measurements were available in 12 patients from the MGMT promoter-methylated group with low NK cell function. Five of these patients showed later on during treatment at least one normal NK cell function result. Normalization of NK cell function was, however, only transient in four patients. Two of the five patients with initially normal NK cell function had later on at least one NK cell function test below the normal level. In the MGMT promoter-unmethylated patients, the low NK cell function normalized in further tests in 8 out of 10 patients with follow-up data, 3 of them, however, showing only a transient normalization. Five out of six patients with normal NK cell function at the first test and with available follow-up data showed a decreased NK cell function later on during treatment.

In the immune-diagnostic blood sampling, and further during treatment, repetitive blood samples were taken to detect CCCs in both the MGMT promoter-methylated and -unmethylated patients. In total, 86 (34 + 52) of the 157 (72 + 87) samples showed the presence of CCCs. There was no statistical difference in the presence of CCCs in both groups.

An important molecule in tumor cells is the expression of PDL1 on its surface. Therefore, the mRNA expression for PDL1 was followed-up in the CCCs in both the MGMT promoter-methylated and -unmethylated patients (Figure 5). At the time of immune-diagnostic blood sampling, only 5 (1 + 4) patients showed mRNA expression for PDL1 out of 27 (14 + 13) samples containing CCCs. However, following mRNA expression for PDL1 in CCCs over time, four more patients in the MGMT promoter-methylated group, and seven more patients in the MGMT promoter-unmethylated group showed positive expression for mRNA for PDL1 in at least one sample.

Similar dynamic changes were observed for the mRNA expression of MGMT (Figure 6). For both MGMT promoter-methylated and -unmethylated patients, a consecutive increase in mRNA expression for MGMT in CCCs was observed in two (p24623 and p25109) and four (p24080, p24084, p24361, p24643) patients, respectively. In only one (p24542) and two (p24084, p24607) patients, respectively, a lower mRNA expression was observed following a high expression in a former sample. Including all samples together, 11/15 (73%) patients in the MGMT promoter-methylated group showed during their disease course at least one sample with high mRNA expression for MGMT in CCCs, when these cells were detectable, compared to 12/16 (75%) patients in the MGMT promoter-unmethylated group.

### 3.4. Survival Outcome of the Patients

Figure 7 shows the data on PFS and OS for both the MGMT promoter-methylated and -unmethylated patients. The date of progression was defined as the date of the MRI that showed signs of progressive disease, diagnosed by the radiologist of the local neuro-oncology clinic, and that led to a change in treatment. There was no reference or central radiological review available. Data from two patients in both the MGMT promoter-methylated and -unmethylated groups were missing. Median PFS for the MGMT promoter-methylated and -unmethylated patients was 27 and 11 months, respectively, with a 12-month PFS of 85% (CI95%: +10, −25) and 43% (CI95%: +19, −20), respectively. Because the ratio of hazard functions (MGMT methylation status) was shown not to be constant over time, the Gehan–Breslow–Wilcoxon test instead of log-rank (Mantel–Cox) test was used to compare the curves. The PFS in both groups was significantly different (*p* = 0.0097). Data on OS were available for all patients. The median OS for MGMT promoter-methylated and -unmethylated patients was 38 and 22 months, respectively, with a 2-year OS of 81% (CI95% +12, −25) and 42% (CI95%: +18, −19), respectively. The difference was again significant (Gehan–Breslow–Wilcoxon test: *p* = 0.0153). The median PFS and median OS of the total group were 17 and 31 months, respectively, with a 12-month PFS of 62% (CI95%: +13, −16; *n* = 46) and a 2-year OS of 59% (CI95%: +13, −16; *n* = 50).

All treatments were provided under the legal frame of “individueller Heilversuch”. In this context, the best possible treatment plan, based on a translation of the most recent insights, was discussed with each patient and was provided after informed consent. Therefore, the treatment plans were modified, patient per patient, and within each patient during the course of the disease over the 6.5-year period. We created a swimmer plot showing the lifespan per patient from operation to death or analysis. The data are shown in Figure 8. In both MGMT promoter-methylated and -unmethylated patients, an improvement in the lifespan of the patients was realized starting from 2018, as compared to the initially treated patients.

### 3.5. Side Effects

All patients were treated in an ambulant setting and retrospectively analyzed. A systematic analysis of all adverse events was not possible. Only a few adverse reactions, possibly or probably related to IMI, were registered in the database. One patient showed hepatobiliary disorders grade 3, linked to the treatment with pembrolizumab. Two patients showed maculopapular skin rash, in one patient most likely linked to TMZ, and in the other patient possibly linked to ICD immunotherapy. General malaise grade 2 was noticed in one patient, who therefore stopped the IMI. Headache and tiredness grade 1 was registered in another patient and was considered possibly linked to the IMI. One patient had a relapse but continued maintenance ICD immunotherapy, in connection to the start of perillyl alcohol inhalations, IV bevacizumab, and increasing dose of prednisolone. Although the patient was assessed as stable for his neuro-oncologic disease, he suddenly showed a massive lethal lung embolism.

### 3.6. The Challenge of Keeping Tumor Control in a Dynamic Cancer Disease

From the above-described combined treatment strategy including IMI, it is clear that the major focus of the immunotherapy part is not dealing with tumor antigen specificity. Indeed, in either the use of tumor lysate or the use of ICD immunotherapy-induced serum-derived antigenic extracellular microvesicles and apoptotic bodies, the tumor-specific neoepitopes are in principle not known. That makes the search and quantification of circulating tumor-specific immune cells induced after vaccination very difficult. However, we were able to demonstrate the proof of principle in a case study [31]. From these data, it is clear that neurosurgery, radiochemotherapy, and TMZ maintenance cycles on their own did not induce a tumor neoantigen-specific T-cell reactivity. However, the integration of 7 ICD immunotherapy courses into seven subsequent 5-day TMZ courses, and administration of one IO-Vac^®^ after chemotherapy, in which antigens derived from ICD immunotherapy-induced antigenic extracellular microvesicles and apoptotic bodies were used for loading the DCs, were able to induce measurable reactivity of both CD4 and CD8 T cells against tumor-specific neoepitopes. This patient was further treated on an individualized basis with repetitive maintenance ICD immunotherapy courses (Figure 9). However, in March 2021, 29 months after the initial diagnosis, she developed an extracerebral intracranial lesion, which, after complete resection, was diagnosed as GBM. Two new IO-Vac^®^ vaccines, loaded with tumor lysate and ICD immunotherapy-induced serum-derived antigenic extracellular microvesicles and apoptotic bodies, were administered together with pembrolizumab. The latter caused some arthritis for which inflammatory drugs and even a short period of low-dose steroids were needed. Nine months after this second event, a diffuse new relapse emerged at the contralateral side. A new incomplete resection was performed, and she was put back on 5-day TMZ chemotherapy cycles in which 5-day immunogenic cell death immunotherapy is integrated. In July 2022, she was still in remission under this treatment. Most interestingly, the antigenic neoepitopes of the second relapse (Table 2) were again analyzed by CeGaT (www.cegat.de (accessed on 1 September 2022)). The tumor was still IDH1 wild-type and MGMT promoter-unmethylated. Instead of 0.5 variants per megabase, the mutational burden was increased to 1.7 variants per megabase. Instead of 10 tumor-specific tumor epitopes, 24 tumor-specific epitopes were predicted in this new tumor, of which only 3 neoepitopes (12.5%) were also predicted in the first tumor. The two neoepitopes to which the CD8 T-cell response was detected were not present in the new lesion anymore. 

## 4. Discussion

A report of a single-center experience on individualized multimodal immunotherapy, integrated within the standard of care, for adults with IDH1wt glioblastoma multiforme is presented. The patient group is an unselected consecutively treated group of adults who met the criteria for being included in the retrospective analysis. The report demonstrates that the conduction of such a complex combined treatment approach is feasible, even when chemotherapy, imaging, and supportive care were performed by the local neuro-oncology clinic while the IMI was administered at the IOZK. The data show again how dynamic a GBM in reality is, including its interaction with the immune system. Nevertheless, the combined treatment approach led to promising PFS and OS. The status of the MGMT promoter methylation, though itself being most likely a dynamic process, still emerged as a prognostic factor. The median OS in the MGMT promoter-methylated patients (38 months) was clearly better than in the unmethylated patients (22 months), with a 2-year OS of 81% and 42%, respectively. Overall, the data illustrate and support the concept that the inclusion of IMI within and after the standard of care for patients with GBM is worth further elaboration.

Since the publication of the randomized controlled trial (RCT) with the inclusion of TMZ during and after radiotherapy [32,33], the standard of care has not been changed. Although not stratified for the MGMT promoter-methylation status, post hoc subanalyses showed the median OS of 23.4 months in the MGMT promoter-methylated patients to be better than the 12.6 months in the MGMT promoter-unmethylated patients, with a 2-year OS of 49% and 15%, respectively [33]. The combined treatment (radiochemotherapy and maintenance TMZ chemotherapy) became the control arm for a new RCT with the addition of TTFs as the experimental arm. Data are available again for both MGMT promoter-methylated and -unmethylated patients [18]. For the MGMT promoter-methylated patients, the OS of the standard arm was slightly less good than in the subanalyses of the former trial, with a median OS of 21 months and a 2-year OS of 38%. This contrasts with the improved OS in the MGMT promoter-unmethylated patients, with a median OS of 15 months and a 2-year OS of 22%. In the two groups, the addition of TTFs increased the median OS to 32 months and 17 months, respectively, with a 2-year OS of 59% and 27%, respectively. The mechanism of how TTFs can improve the OS in combination with TMZ is still a matter of debate. The application of TTFs was originally pointed to a cell cycle arrest of tumor cells [34,35], but such a working mechanism might block the activity of TMZ, which relies on the subsequent cell division of cells having DNA affected by alkylating agents. Nowadays, it is recognized that TTFs induce ICD [19] and activate STING and IM2 inflammasomes [36] to induce adjuvant immunity. Another interesting phase III placebo-controlled randomized trial with the addition of DCVax^®^-L to the standard of care was recently published [37]. This report gives more details than the former report by the same group [38]. The original primary endpoint of the randomized patients was the PFS determined by magnetic resonance. After recruitment, this endpoint had been changed toward OS, and an external control arm was designed. The study did not take into account the MGMT promoter-methylation status for randomization. Data on OS in the experimental arm covered 232 patients randomized to initial DCVax^®^-L treatment. In this group of patients, data on 90 patients with MGMT promoter-methylated status showed a significantly better median OS of 30 months in comparison to the mOS of 21 months in the external control arm. These data are very comparative to the data reached with TTFs [18]. Data on MGMT promoter-unmethylated patients were available in the supplementary data. The data on the median OS of 14.9 months with a 2-year OS of 19% were not different from the median OS of 14.6 months with a 2-year OS of 21% in the external control arm. With the combination of ICD immunotherapy, active specific immunotherapy, and modulatory immunotherapy, the presented data from the current retrospective analysis show a further improvement of the OS. Although significance has not been calculated, and the level of evidence is low because of the retrospective analysis, the improved OS data have high relevance for the patients.

We designed the combined treatment strategy in 3 phases.

1. The first phase is an anticancer strategy in which the maintenance chemotherapy is strengthened with the ICD immunotherapy. Several arguments have been proposed in the past to add the vaccine strategy already after the radiochemotherapy, hence prior to the start of and during maintenance chemotherapy. Nevertheless, data from the RCT HGG-2010 (EudraCT 2009-018228-14) suggested that DC vaccination after maintenance chemotherapy resulted in a higher 2-year OS for both the complete and the less-than-complete resected patients [39]. In order to avoid a monotherapeutic phase during the maintenance TMZ treatment, we included ICD immunotherapy with the combination of injections of Newcastle disease virus and sessions of modulated electrohyperthermia. The mechanisms for killing tumor cells are different, and supposed to be at least additive. Moreover, the components of the ICD immunotherapy were also aimed to change the tumor microenvironment [40], already in preparation for the second phase of the combined treatment.

2. The second phase consists of active specific immunotherapy using DC vaccines. We opted to use freshly isolated monocytes for the production of DCs, thereby avoiding the injection of frozen/thawed DC vaccines. This strategy is different from most actual DC vaccine trials. However, another difference might be much more important. In almost all patients, we used ICD immunotherapy-induced serum-derived antigenic extracellular microvesicles as a source of tumor antigen. ICD immunotherapy was administered during each DC culture period. With this approach, we have demonstrated that we were able to induce immune responses against tumor-specific neoepitopes [31]. Because radiotherapy and TMZ might change the mutational burden in tumor cells [41,42], the antigenicity might change as well. Therefore, the use of tumor lysate of the resected tumor, usually 8 months before the start of DC vaccination, was considered less appropriate. Of note, the potential clinical efficacy of the combination of ICD immunotherapy, together with active specific immunotherapy with DC vaccines, was illustrated in the report by Liau et al. [37]. The addition of TTFs as an ICD inducer to the DC vaccination at the time of relapse resulted in an unexpectedly long OS. Overall, the combination of several modes of immunotherapy becomes broadly accepted [43].

3. The third phase is aimed to maintain and broaden the immune protection through repetitive sessions of ICD immunotherapy. Potential new tumor clones are aimed to be attacked by the NDV and the modulated electrohyperthermia so that an immune response can be induced.

The patient group presented in this report was retrospectively created according to a priori-defined criteria, without further selection. The patient population consists of adults with GBM, defined by the 2016 classification [44] and the 2021 classification [45]. We could compare the patient characteristics with those of the 1366 patients treated in the standard arms of the large phase III RCTs [18,46,47,48,49] that were used for creating the external control arm for the DCVax^®^-L trial [37]. Our patient group was slightly younger, with a median age of 48 years, whereas the median age in the reports ranged from >50 to 60 years. In our cohort, we had a female/male ratio of 46%/54%, in comparison to ranges from 28%/72% to 42%/58% in the literature. The median KPI of 80 was lower than the median KPI of ≥90 in most trials (except the study from Weller et al., who published a median KPI above 70). With a percentage of 56% MGMT promoter-unmethylated patients, this distribution fell within the reported range, going from 51 to 69%. Gross total resection was present in 46% to 74% of the reported patients, while in our cohort, only 28% of patients were marked as R0 and 52% as <R0. Besides the MGMT promoter methylation status, the extent of resection remains a significant prognostic factor for OS [50]. Overall, we can conclude that the prognostic factors present in our patient group were in line with the patient groups reported in the literature.

The presence of primary malignant brain tumors elicits immune defects [51], which recover after resection [52]. Most of the blood analyses presented here, however, were taken after the radiochemotherapy. The stronger proportion of patients with low total white blood cells is higher in the MGMT promoter-methylated group, most likely due to more sensitivity to chemotherapy. The low number of the lymphocytic compartment in both MGMT promoter-methylated and -unmethylated patients fits with data published in the literature [53]. In both groups, we found a higher proportion of patients with IL-17 skewing, which could be corrected during treatment, especially in the MGMT promoter-methylated patients. The role of IL-17 in glioma progression has been described [54], acting both at the side of the glioblastoma cells [55] and at this side of the immune function [56]. Intermediate levels of IL-17 were correlative to favorable OS in a personalized peptide vaccination trial [57]. The high proportion of patients with low NK cell function in both patient groups remained high during treatment. This is an important observation, and new strategies have to be developed to increase the NK cell function [58,59]. Earlier studies indeed showed the role of the NK cell response associated with improved survival in patients [60,61].

Liquid biopsy is an important novel medical tool in the treatment of GBM patients. First of all, it allowed us to yield actual present tumor antigens out of the serum upon induction with ICD immunotherapy. With these antigens loaded on DCs, we demonstrated the induction of a tumor neoantigen-specific immune response [31]. The follow-up of this patient, however, showed another critical challenge. A new neoantigen analysis on a relapsed tumor, 38 months later, showed only three remaining tumor antigens. The two neoantigens that were targeted by the CD8+ T cells had disappeared, and an increase in tumor mutational burden with much more novel tumor antigens was detected. Similar downregulation of immune-targeted antigens at the time of relapse has been observed [62]. The persistence of ICD immunotherapy courses for a long period, aimed to kill newly developing subclones and to include their antigenicity within the global immune protection induced by the IMI, might therefore be a crucial element in the combined treatment approach. Further molecular analyses on ICD immunotherapy-induced circulating tumor DNA, EVs, and apoptotic bodies are needed to elaborate on this point and develop necessary tools for the early detection of tumor relapse by novel escaping cancer cell subclones, and for the early detection of changing tumor–host interactions.

Another potential strategy to follow the tumor biology in a noninvasive way is the analysis of CCCs in blood samples over time. The existence of these cells in patients with GBM has been described [63,64]. The analysis of all our patient samples was performed at Biofocus, and was based on a four-marker assay [65]. For carcinoma, this test reaches a sensitivity and specificity of 80 and 96%, respectively. According to the company, CCCs are detected in about 60% of samples from brain tumor patients. The detection is based on the mRNA expression, measured with quantitative real-time PCR, for the markers telomerase, ERBB2, C-KIT, and EGFR, in cells isolated using a filtration-based technique. No data on sensitivity and specificity are provided by the company. The coefficients of variance for the intra- and interassay measurements reported by the company are <1.3 and <2.2%, respectively. Taking all blood samples together in this reported cohort of patients, CCCs were detected in 55% of the samples. Once detected, the mRNA expression for PDL1 and for MGMT could be determined and followed over time. Intra- and interassay variation reported by the company is <1% and <2.7%, respectively. Reproducible objective detection of PDL1 expression in tumor tissue has been a challenge in the past, because of a multitude of detection methods, antibodies, staining protocols, readout methods, and cutoff definitions [66]. On top of that, the expression within a tumor is heterogeneous [67]. Similar to conclusions from pathology analyses, a continuous variable of the ratio of mRNA expression of the marker of interest in comparison to the GAPDH mRNA expression in CCCs was finally categorized toward high or low, with a cutoff of 2. Blood sampling challenges might be present as well, though the liquid biopsy might reflect better the global status of the cancer disease. PDL1 expression might fluctuate with the cell cycle of GBM cells [68]. In addition, the presence of interferon-gamma might influence the PDL1 expression [69]. PDL1 expression might differ between primary and recurrent GBM [70]. These all explain why PDL1 expression on tumor cells might change over time. We believe that the detection of mRNA for PDL1 on CCCs might reflect the dynamic changes occurring in the tumor over time, and might indicate the time point to eventually add checkpoint inhibitors as modulatory immunotherapy. Similar diagnostic challenges exist for the detection and categorization of the MGMT promoter methylation status in tumor tissue [71]. A large number of techniques have been used, and there is no consensus on any cutoff in the categorization. It is generally accepted that the MGMT promoter methylation status remains more or less stable between primary and recurrent GBM [72,73]. Nevertheless, we found that some patients had CCCs over time in which the mRNA expression for MGMT raised and reached the cutoff value. It is not excluded that the relative amount of MGMT promoter-unmethylated subclones from the tumor at certain time points become more reflected within the CCCs. Because the MGMT promoter methylation status is a prognostic factor [72], the subtle changes within the CCCs might reflect the impact over time of MGMT promoter-unmethylated subclones on the treatment efficacy with alkylating agents.

Facing a bad prognosis of the disease, the frequency and intensity of adverse reactions should be taken into account when any innovative treatment is introduced. From a recent meta-analysis of available RCTs, the addition of DC vaccines in the treatment of GBM did significantly prolong the OS, without increasing the incidence of adverse events [74]. In the phase III RCT from Liau et al., the frequency and intensity of adverse events related to vaccination were low [37]. Except for flu-like symptoms being the most commonly reported adverse reaction after intravenous NDV administration, no adverse reactions are reported in the literature [75]. The application of modulated electrohyperthermia to brain tumors has also been published as a treatment with only mild adverse reactions [20,76]. Because our patients were individually treated in an outpatient setting and outside any prospective study protocol, we most likely could not register all potential adverse reactions. However, the spectrum of registered adverse reactions is in line with the literature. Overall, our clinical experience with IMI fit with the published analysis of side effects induced by cancer vaccines and oncolytic viruses in comparison to other systemic anti-cancer therapies [77].

Our study has limitations. Although the profile of patients for sampling data out of the database was a priori defined, it remains a retrospective analysis, and comparison with the control arms of RCTs or external control arms can only reach low evidence for higher efficacy of IMI included in first-line treatment for the improvement of OS. The particular challenges for randomized controlled immunotherapy clinical trials for GBM have been published [31].

The individualized treatment approach presented here allows a continuous evolution of combined treatment strategies. This evolution occurs at two levels. First of all, fixing study protocols for many years into a clinical trial blocks early translation of relevant novel scientific insights into clinical reality during the complete inclusion period. Sticking to fixed study protocols, which might have become outdated in the light of the fast-evolving scientific knowledge, is of benefit for the clinical research, but might at the same time strongly affect potential benefits for new patients. A second level for the evolution of combined treatment strategies occurs within each patient. We have adapted the treatment during treatment, based upon monitored biological changes in the tumor and tumor–host interaction, because we were convinced that dynamic tumors should not be treated with fixed protocols. Besides adapting treatment strategies from patient to patient, and within the patient, another issue emerges. The real-world data presented in this analysis have a distinct scientific and clinical value for patients. Due to strict inclusion and exclusion criteria, such a value can hardly be reached in clinical trials, because patients considered eligible for phase III clinical trials represent a highly selected minority of patients in a real-world GBM population [78]. This all makes that the paradigm of clinical research for GBM has to change. With our approach, the improvement of outcomes for the patients became clear. Therefore, the relevance of our data for future real-life patients is high. Further work on the cost-effectiveness and determination of the gain of quality-adjusted life years (QALY) is in plan to convince the community about the real value of IMI as an essential component of the standard of care for patients with IDH1wt GBM.

## 5. Conclusions

Single institution real-world data on adults with IDH1wt GBM and treated with individualized multimodal immunotherapy as part of the first-line treatment are summarized in a retrospective analysis. The ambulant treatment is feasible without additive toxicity. The data illustrate that GBM tumors are dynamic biologic processes. Patients might benefit from protocol adaptations during treatment rather than fixed treatment protocols. Liquid biopsy might become an important instrument for repetitive monitoring of the tumor characteristics and the tumor–host interaction. The data on OS are relevant for patients and should be further explored.

## Figures and Tables

**Figure 1 cancers-15-01194-f001:**
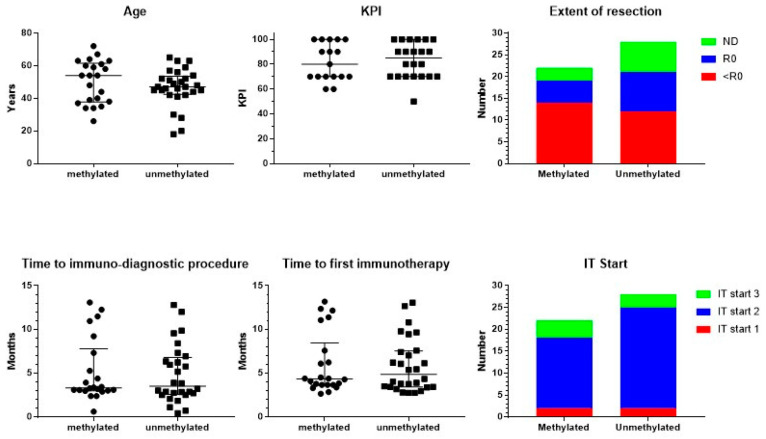
Patient characteristics. Patients were divided into those suffering from MGMT promoter-methylated (*n* = 22) or -unmethylated (*n* = 28) GBM. The individual distribution of age and Karnofsky performance index (KPI), including median and interquartile range, is shown. The figure on extent of resection includes the number of patients with complete resection (R0), biopsy or noncomplete resection (<R0), and not documented (ND). In the lower panel, time in months from neurosurgery to the immune-diagnostic procedure at the IOZK, and to the start of the first immunotherapy at the IOZK, for both groups of patients is shown, including median and interquartile range. In the lower panel right figure, the patients are categorized according to the start of the immunotherapy (IT Start): patients who started immunotherapy without additional chemotherapy (IT Start 1), patients who started ICD immunotherapy during the temozolomide maintenance chemotherapy (IT Start 2), and patients who started individualized multimodal immunotherapy after maintenance chemotherapy (IT Start 3).

**Figure 2 cancers-15-01194-f002:**
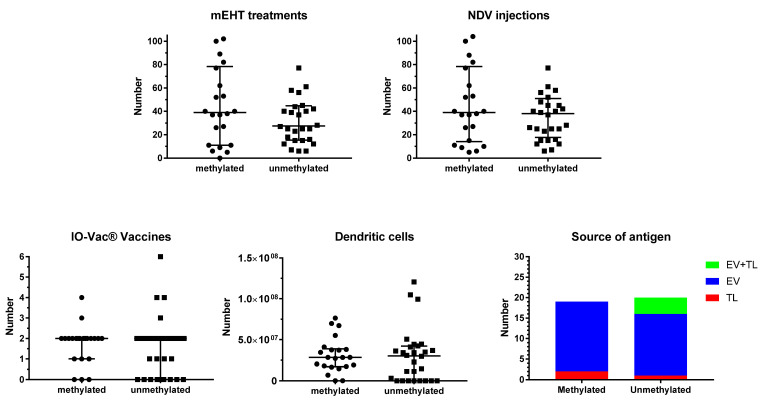
Treatment details. Patients were divided into those suffering from MGMT promoter-methylated (*n* = 22) or -unmethylated (*n* = 28) GBM. The individual distribution of number of sessions of modulated electrohyperthermia (mEHT) and injections of Newcastle disease virus (NDV) are shown in the upper panel, including median and interquartile distribution. In the lower panel, the total number of IO-Vac^®^ vaccines and the total number of dendritic cells injected per patient are shown, including median and interquartile distribution. In the right lower panel right, the source of antigen for loading into the DC vaccine (IO-Vac^®^) for each patient, who received IO-Vac^®^, is shown: TL = tumor lysate; EV = extracellular vesicles: Immunogenic cell death-induced serum-derived antigenic extracellular microvesicles and apoptotic bodies; EV + TL = combination of both sources of antigen.

**Figure 3 cancers-15-01194-f003:**
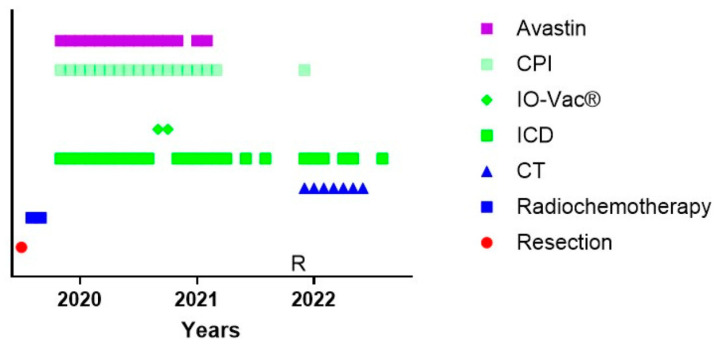
Treatment schedule of a female suffering from GBM in the context of POLE syndrome. The treatment schedule of a female with POLE syndrome is shown. CT = maintenance temozolomide; ICD = immunogenic cell death immunotherapy; IO-Vac^®^ = dendritic cell vaccine; CPI = checkpoint inhibitor anti-PD1 monoclonal antibody; R = multifocal relapse.

**Figure 4 cancers-15-01194-f004:**
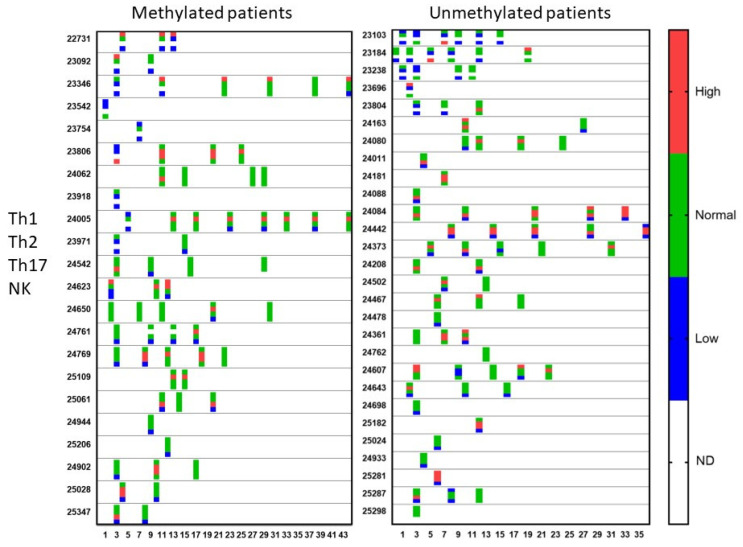
Qualitative immune function during treatment. Patients were divided into those suffering from MGMT promoter-methylated (*n* = 22) or -unmethylated (*n* = 28) GBM. At different time points during treatment (expressed in X-axis as months), the immune functioning was followed for the Th1 function, Th2 function, Th17 function, and NK cell cytotoxicity. The data are categorized as higher or lower than or within (normal) the reference range, determined by the laboratory. For each patient and for each test in the time, the four dots for the four tests with the respective color are shown. ND = test not done.

**Figure 5 cancers-15-01194-f005:**
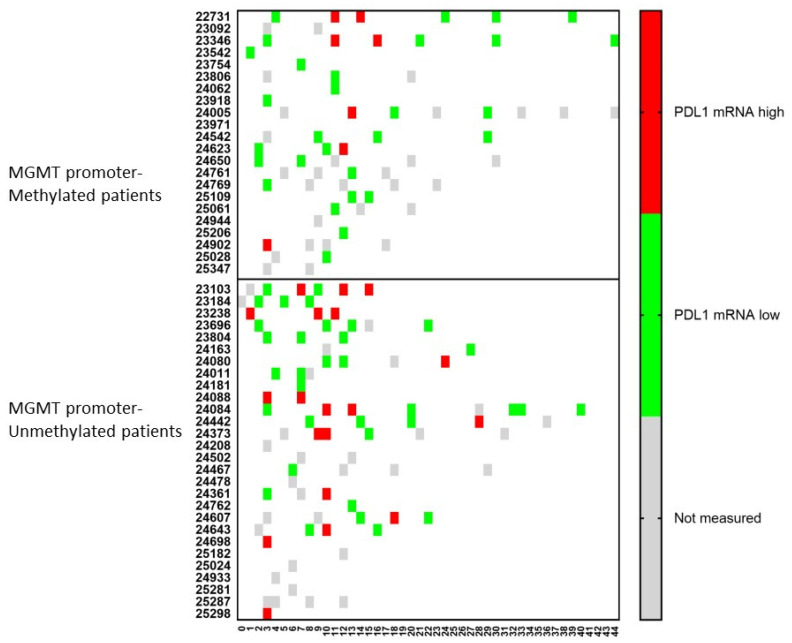
The relative amount of PDL1 mRNA expression over time in circulating cancer cells. Patients were divided into those suffering from MGMT promoter-methylated (*n* = 22) or -unmethylated (*n* = 28) GBM. At different time points during treatment (expressed in X-axis as months), the CCCs were determined, and in case of detection, the mRNA for PDL1 was determined. The expression was high or low when the ratio of mRNA expression of PDL1 to GAPDH was more or less than 2. Not measured: No CCCs were detected, or PDL1 mRNA expression was not determined.

**Figure 6 cancers-15-01194-f006:**
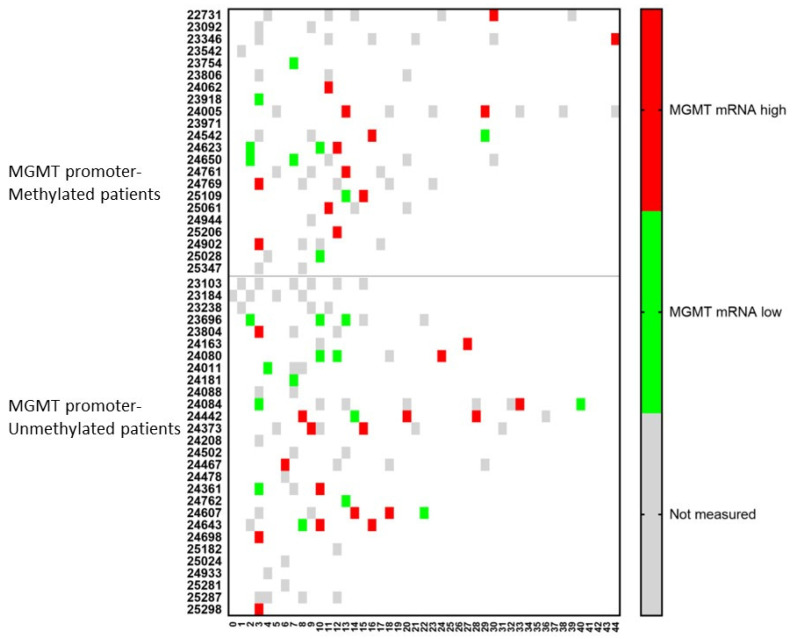
The relative amount of MGMT mRNA expression over time in circulating cancer cells. Patients were divided into those suffering from MGMT promoter-methylated (*n* = 22) or -unmethylated (*n* = 28) GBM. At different time points during treatment (expressed in X-axis as months), the CCCs were determined, and in case of detection, the mRNA for MGMT was determined. The expression was high or low when the ratio of mRNA expression of MGMT to GAPDH was more or less than 2. Not measured: no CCCs were detected, or MGMT mRNA expression was not determined.

**Figure 7 cancers-15-01194-f007:**
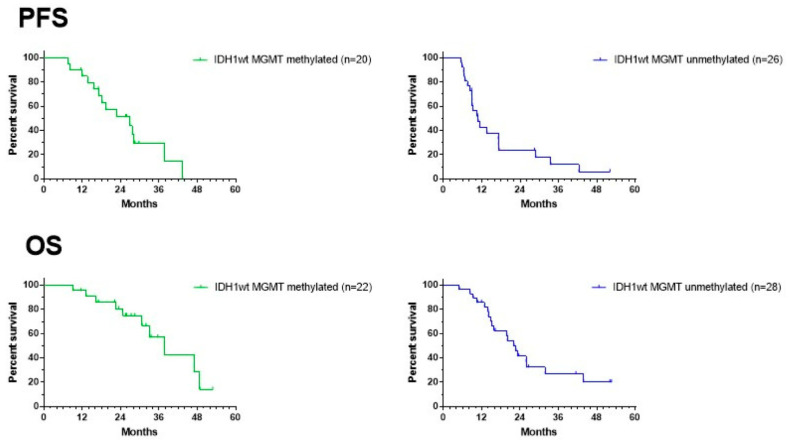
Survival curves. PFS (upper panel) and OS (lower panel) of the groups of MGMT promoter-methylated (left) and -unmethylated GBM patients. For PFS, two patients are missing in each group of patients.

**Figure 8 cancers-15-01194-f008:**
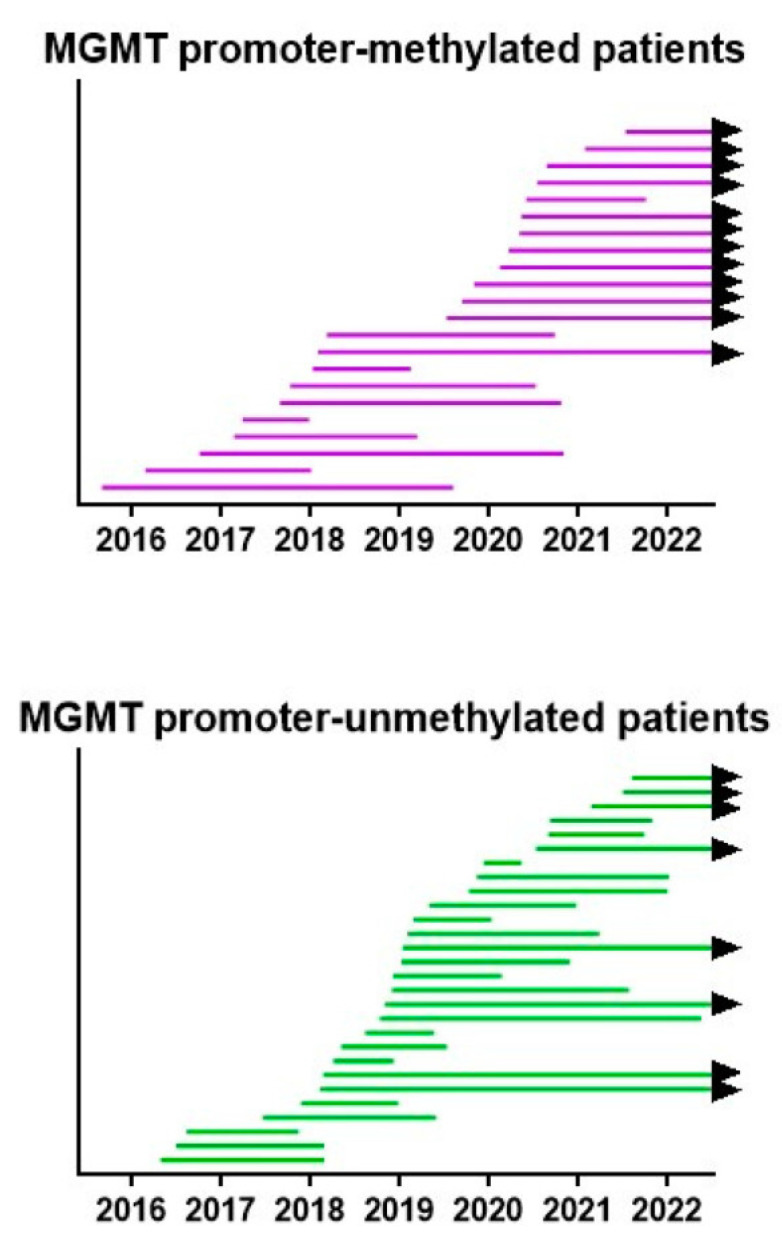
Survival swimmer plot. The lifespan between neurosurgery and death, or 1 July 2022 when the patient is still alive (black arrow), is shown for both MGMT promoter-methylated and -unmethylated patients. Patients are sorted according to the date of neurosurgery.

**Figure 9 cancers-15-01194-f009:**
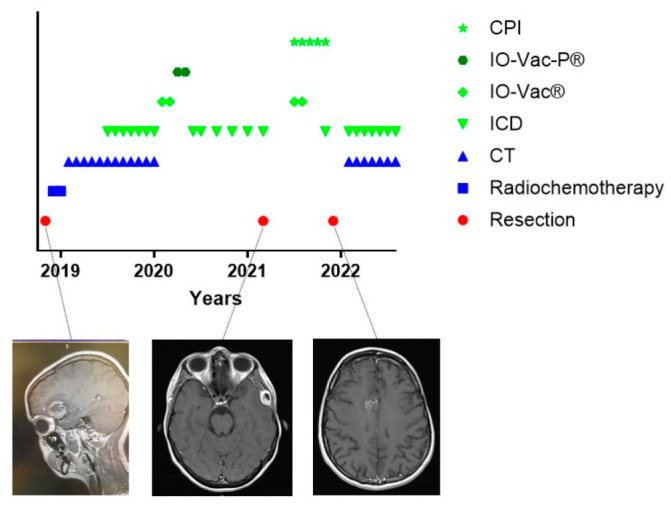
Treatment schedule of a female suffering from relapsing GBM. The treatment schedule of a female patient with GBM. The treatment for the first event of this patient was published [31]. Further evolution with two new relapses is shown. The original tumor (**left**), the extracerebral intracranial first relapse (**middle**), and the diffuse relapse at the contralateral side (**right**) are shown in exemplary MRI images. The predicted neoepitopes in the first tumor and last tumors are described in Table 2. CT = maintenance Temozolomide; ICD = immunogenic cell death immunotherapy; IO-Vac^®^ = dendritic cell vaccine; IO-Vac-P^®^ = dendritic cell vaccine loaded with tumor-specific neoepitopes; CPI = checkpoint inhibitor anti-PD1 monoclonal antibody.

**Table 1 cancers-15-01194-t001:** Blood analysis at time of immune-diagnostic procedures.

	Methylated			Unmethylated			Chi-Square
	L (%)	N (%)	H (%)	L (%)	N (%)	H (%)	
White Blood cell count	6 (27)	15 (68)	1 (5)	1 (4)	23 (82)	4 (14)	**0.0402**
Lymphocytes	16 (73)	5 (23)	1 (5)	16 (59)	11 (41)	0 (0)	0.2505
Monocytes	6 (27)	15 (68)	1 (5)	3 (11)	24 (89)	0 (0)	0.1649
Neutrophils	4 (18)	15 (68)	3 (14)	1 (4)	21 (78)	5 (19)	0.2442
Eosinophils	4 (18)	17 (77)	1 (5)	2 (7)	24 (89)	1 (4)	0.5051
Basophils	1 (5)	21 (95)	0 (0)	0 (0)	25 (93)	2 (7)	0.2384
Hemoglobin	5 (24)	16 (76)	0 (0)	3 (11)	23 (82)	2 (7)	0.2449
Platelets	9 (41)	12 (55)	1 (5)	6 (21)	22 (79)	0 (0)	0.1439
T cells	18 (82)	3 (14)	1 (5)	19 (73)	7 (27)	0 (0)	0.3151
B cells	18 (82)	4 (18)	0 (0)	20 (27)	5 (19)	1 (4)	0.6411
NK cells	15 (68)	7 (32)	0 (0)	17 (65)	9 (35)	0 (0)	0.8377
Th2 skewing	5 (23)	15 (68)	2 (9)	3 (11)	16 (57)	9 (32)	0.1148
Th17 skewing	1 (8)	7 (54)	5 (38)	0 (0)	13 (57)	10 (43)	0.4005
Th1 skewing	4 (18)	15 (68)	3 (14)	1 (4)	23 (82)	4 (14)	0.2288
Th1/Th2	3 (23)	9 (69)	1 (8)	8 (33)	15 (63)	1 (4)	0.7591
NK cell function	16 (73)	5 (23)	1 (5)	18 (67)	9 (33)	0 (0)	0.3312

	**Good (%)**	**Moderate (%)**	**Increased (%)**	**Good (%)**	**Moderate (%)**	**Increased (%)**	
Oxidative load	5 (24)	9 (43)	7 (33)	7 (28)	7 (28)	11 (44)	0.5675
	**Low (%)**	**Medium (%)**	**High (%)**	**Low (%)**	**Medium (%)**	**High (%)**	
Total anti-oxidative capacity	3 (14)	5 (24)	13 (62)	2 (8)	3 (12)	20 (80)	0.3963

	**No CCC (%)**	**CCC PDL1− (%)**	**CCC PDL1+ (%)**	**No CCC (%)**	**CCC PDL1− (%)**	**CCC PDL1+ (%)**	
CCC	7 (33)	13 (62)	1 (5)	13 (50)	9 (35)	4 (15)	0.1467

CCCs = circulating cancer cells; PDL1− and PDL1+ = the ratio of mRNA for PDL1 in comparison to the mRNA for GADPH not higher and higher, respectively, than the cutoff of 2; Th = T helper.

**Table 2 cancers-15-01194-t002:** Neoepitopes predicted in first event and second relapse of patient with GBM.

	October 2018 (Primary Tumor)			December 2021 (Second Relapse)	
No	Peptide	NAF (DNA)	No	Peptide	NAF (DNA)
1	DLKNRTGFAV	0.46	11	RLASDLAEF	0.63
2	SLHNHMRFR	0.26	12	FAARPCAEI	0.61
3	HFFCDTYPLLK	0.43	13	IMENSPKDVY	0.33
4	RIFNLISM	0.16	14	RVALVPIKY	0.26
5	ALDIRAHIEEF	0.25	15	NCNGPSPNM	0.4
6	KVHQNIHTGEK	0.17	16	GSHGYDLSTF	0.76
7	KAGLKVHQNIHTGEKPH	0.09	17	RSDHYSEEL	0.19
8	RGANPDLKNRTGFAVIH	0.46	18	PAAPYIPGL	0.35
9	TCPLPSSLHNHMRFRHS	0.26	2	SLHNHMRFR	0.44
10	VALDIRAHIEEFKPYI	0.25	19	SVSAPAFYSPQK	0.09
			20	KTSYIIMIGPD	0.19
			21	YEVLLVTSSFVSPSESRSG	0.61
			8	RGANPDLKNRTGFAVIH	0.4
			22	IMENSPKDVYVVQIEAFD	0.33
			23	TPYLLHFSNVSVPRVRAE	0.34
			24	AEPEKMGGDGTVCSPLE	0.43
			25	SVESGANDVVFIRTLG	0.29
			26	PKRGSEGGLAAFVDFVD	0.17
			7	KAGLKVHQNIHTGEKPH	0.61
			27	AKQESLETLVLSGIGST	0.3
			28	KSNHDKNVTPDEVLQTL	0.18
			29	FSQKSRVTENPTEALS	0.31
			30	AEPPGTPPDSHSHLDAA	0.34
			31	AISWARTKRIPFLGV	0.08

NAF: Novel allele frequency. Peptides marked in green (**left**): A CD8+ T-cell response was detected against neoepitope 5, 6, and 7 (dark green), and a CD4+ T-cell response against neoepitope 8 (light green) upon individualized multimodal immunotherapy. Peptides marked in yellow (**right**): Neoepitopes that were present in the first tumor and still present in the second relapse.

## Data Availability

The data presented in this study are available on request from the corresponding author. The data are not publicly available due to privacy reasons.

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
