# Peer review of "Individualized Multimodal Immunotherapy for Adults with IDH1 Wild-Type GBM: A Single Institute Experience"

_cancers, 2023, doi:10.3390/cancers15041194_

Round 1

Reviewer 1 Report

Individualized multimodal immunotherapy improves overall survival of adults with IDH1 wild-type GBM

The current manuscript describes the retrospective analysis of a group of adults taken from a database following a predefined clinical profile without any bias. It discusses about the overall survival of the of patients after multimodal immunotherapy along with the chemotherapy. The proposed treatment concept showed the dynamic changes in the tumor biology and tumor host interaction that proposes an additional paradigm of protocol medicines in clinical trials. The important message was that there was no major treatment related adverse reactions.

The authors conclude that multimodal immunotherapy improves the overall survival of adults with IDH1 wild type glioblastoma.  

The manuscript in current form is very well written and have no further comments

Author Response

We thank the reviewer for reading and analyzing the manuscript, and for approval of the content of the manuscript. 

Reviewer 2 Report

The authors present a cohort of glioblastoma patients treated as a part of the initial maintenance treatment with TMZ. They report favorable overall survival in these patients compared to historical controls. The intervention appears safe. They coupled liquid biopsy of circulating cancer cells during treatment although the implications of these data are unclear. Variation in detection of the cells then variation in expression levels of MGMT and PDL1 may reflect tumor biology, treatment effect, or technical variation. 

The title is not supported by the data. With the current evidence the most assertive claims would be that the treatment provided was well tolerated. This report does not support efficacy claims. Further rigorous study in a prospective randomized fashion would be needed. The conclusions are more reasonably stated although the sentence on line 777-778 is overstated as well. The data suggest patient might benefit from adaptive treatment. There is no definitive evidence presented that they do benefit.

Author Response

We thank the reviewer for reading the manuscript, reflecting on the data and making valuable proposals to improve the accuracy of the manuscript.

The title is not supported by the data. With the current evidence the most assertive claims would be that the treatment provided was well tolerated. This report does not support efficacy claims. Further rigorous study in a prospective randomized fashion would be needed.

We have changed the title from “Individualized multimodal immunotherapy improves overall survival of adults with IDH1 wild-type GBM” towards “Individualized multimodal immunotherapy for adults with IDH1 wild-type GBM: a single institute experience”.

The conclusions are more reasonably stated though the sentence on line 777-778 is overstated as well. The data suggest patient might benefit from adaptive treatment. There is no definitive evidence presented that they do benefit.

We have changed the sentence “The data illustrate that GBM tumors are dynamic biologic processes which require treatment adaptations instead of fixed treatment protocols” towards “The data illustrate that GBM tumors are dynamic biologic processes. Patients might benefit from protocol adaptations during treatment rather than fixed treatment protocols.